# Calcium Pyrophosphate and Basic Calcium Phosphate Crystal Arthritis: 2023 in Review

**Augustin Latourte [1,2,*]** , **Hang-Korng Ea [1,2]** and **Pascal Richette [1,2]**

1   INSERM, UMR-S 1132 BIOSCAR, Université Paris Cité, F-75010 Paris, France
2   Rheumatology Department, AP-HP.Nord, Lariboisière Hospital, F-75010 Paris, France
*   Correspondence: augustin.latourte@aphp.fr

**Abstract:** Calcium-containing crystal deposition diseases are extremely common in rheumatology. However, they are under-explored compared to gout or other inflammatory rheumatic diseases. Major advances have been made in 2023 that will undoubtedly stimulate and facilitate research in the field of calcium pyrophosphate (CPP) deposition disease (CPPD): the ACR/EULAR classification criteria for CPPD and a semi-quantitative OMERACT score for ultrasound assessment of the extent of CPP deposition have been validated and published. A large randomized controlled trial compared the efficacy and safety of colchicine and prednisone in acute CPP arthritis. Preclinical studies have elucidated the pro-inflammatory and anti-catabolic effects of basic calcium phosphate (BCP) crystals on mononuclear cells and chondrocytes. The association between osteoarthritis (OA) and IA calcifications has been the subject of several epidemiological publications, suggesting that calcium crystals are associated with a greater risk of progression of knee OA. Research in the field of calcium crystal deposition diseases is active: the areas of investigation for the coming years are broad and promising.

**Keywords:** CPPD; BCP; classification criteria; colchicine; prednisone; imaging; osteoarthritis

## 1. Introduction

Calcium pyrophosphate (CPP) and basic calcium phosphate (BCP) crystals are the two main types of calcium crystals responsible for clinical manifestations in and around the joints [1]. The conditions associated with these crystals are extremely common in everyday practice but have long been relatively neglected by clinical and basic research, with the result that there is currently no specific treatment to manage them.

Nevertheless, calcium crystal-associated diseases have recently received renewed attention and several works have been the subject of major publications in 2023: these publications were presented at the G-CAN Congress in November 2023 in La Jolla, CA. The aim of this review is to provide an update on recent advances in the field of CPPD and BPC-associated disease.

## 2. CPP Crystals

### 2.1. ACR/EULAR 2023 Classification Criteria

The year 2023 was marked by the publication of the first international classification criteria for CPPD, an initiative supported by the ACR and EULAR [2]. The aim of the classification criteria is to identify relatively homogeneous patient groups, thereby facilitating epidemiological research, clinical trials, and basic research.

These classification criteria concern only symptomatic forms of CPPD but their ambition was to cover all manifestations of this highly heterogeneous rheumatism, namely acute CPP arthritis, chronic inflammatory CPP arthritis, and OA associated with CPPD [3]. They apply to any patient who has joint pain, swelling, or tenderness and whose overall symptoms are not explained by another pathology (which would be an exclusion criterion).

It is therefore possible for a patient with rheumatoid arthritis or gout to be classified as having CPPD if some of their symptoms are better explained by CPPD than by their other rheumatic condition.

Two criteria are considered "sufficient" to immediately classify a patient as having CPPD: identification of CPP crystals by synovial fluid analysis or histology and crowned dens syndrome. In the absence of these sufficient criteria, classification depends on weighted relative criteria, the sum of which must be > 56 points to classify a patient as having CPPD. These relative criteria are divided into eight domains, three of which are imaging domains: the presence of features of hand/wrist OA on imaging (domain F); imaging evidence of CPPD in the symptomatic peripheral joint(s) (domain G); and number of peripheral joints with imaging evidence of CPPD (domain H). Clinical domains include age at symptom onset (domain A); features of inflammatory arthritis (domain B); peripheral joints affected by $\geq 1$ typical episode of acute CPP arthritis (domain C); and the presence of associated metabolic disease (domain D). The maximal score is 35 for the clinical domains and 48 for the imaging domains. Overall, it is impossible to classify a patient as having CPPD in the absence of suggestive clinical manifestations and/or in the absence of suggestive calcium deposits on imaging.

Finally, it should be noted that some items are negatively weighted in these classification criteria: involvement of the first MTP (considered more suggestive of gout than CPPD), absence of CPP crystals on analysis of joint fluid by polarized light microscopy (*a fortiori* when several attempts have been made to identify the crystals; domain E), and absence of deposits on so-called "advanced" imaging (ultrasound, CT, or DECT, which are more sensitive imaging modalities than standard radiography for detecting intra-articular [IA] calcium deposits) [4].

The ACR/EULAR classification criteria are expected to lead to major advances in clinical and basic research, opening up exciting prospects in the field of CPPD [2]. For example, by facilitating patient recruitment, these criteria should make it easier to conduct clinical trials, particularly in the inflammatory forms of CPPD. These criteria will also allow the constitution of large homogeneous cohorts of patients and epidemiological studies could provide a better understanding of the natural history of the disease.

*2.2. Imaging*

In parallel with the development of the ACR/EULAR classification criteria, an imaging working group has proposed consensus definitions for imaging criteria [5]. These definitions cover the demonstration of calcium deposits suggestive of CPPD on conventional radiography, CT, or DECT. Definitions for ultrasound had already been proposed by the OMERACT working group [6].

The main purpose of these definitions is to help differentiate between calcifications suggestive of CPP, which are generally linear or punctate and less dense than cortical bone, and calcifications suggestive of BCP, which are similar in density to cortical bone. These deposits can be found in fibrocartilage or hyaline cartilage, synovial membrane or joint capsule, ligaments, and tendons.

The performance of these new definitions for conventional radiography was evaluated in an international study of 67 patients with knee osteoarthritis eligible for total knee replacement, the majority of whom (*n* = 31) had Kellgren and Lawrence grade 3 osteoarthritis [7]. The diagnostic performance of knee radiographs performed in the 6 months prior to surgery was estimated using histological identification of CPP crystals in the surgical specimen as the gold standard.

In this study, radiographs were read by two radiologists, an experienced rheumatologist and a junior rheumatologist. Overall, the inter-reader agreement (kappa values) was substantial to almost perfect for the radiologists and moderate to substantial for the rheumatologists. In this study, the specificity of radiography was very good at 92% but the sensitivity was poorer at 54%. The diagnostic accuracy of radiography in detecting chondrocalcinosis was 73%.

The proposed new definitions have therefore not changed the diagnostic performance of standard radiography for the detection of chondrocalcinosis [4]. Ultrasound (US), which has similar specificity but greater sensitivity, is an interesting tool for the diagnosis of calcium deposits suggestive of CPP [4].

In addition to its diagnostic properties, the US may be useful in quantifying the extent of CPP deposits. The OMERACT working group has developed and validated a semi-quantitative grading system (grades 0–3) for US-detected CPP deposits at three different sites: the hyaline cartilage and meniscus at the knee and the TFCC at the wrist [8]. These sites are the most affected by CPP deposition and the most accessible by US [9,10]. The intra- and inter-observer reliability of this scoring system was substantial when all three sites were considered but appeared to be significantly better at the knee (substantial at the meniscus and almost perfect at the hyaline cartilage) than at the wrist (moderate).

This scoring system has many potential applications: it could be used to follow the evolution of deposits over time in the same patient and to monitor the effectiveness of a hypothetical treatment aiming to dissolve CPP deposits, for example.

Another advanced imaging modality particularly studied is CT, which has a higher sensitivity than standard radiography for detecting calcium deposits. In the MOST cohort, which included 2070 patients with or at risk of knee osteoarthritis (mean age [SD] 61.1 [9.6] years, 56.7% female, mean BMI 28.8 [5.2] $kg/m^2$), radiographic chondrocalcinosis was detected in 6.8% of patients, compared with 12.9% on CT [11]. IA mineralization was observed on CT in 5.2% of knees without radiographic chondrocalcinosis. In most cases, calcifications were observed in all three compartments analyzed (medial femorotibial, lateral, and patellofemoral).

### 2.3. Therapeutics

In the absence of treatments able to dissolve the CPP crystals, the management of CPPD focuses on treating microcrystalline inflammation. It is modeled on the treatment of gout and relies primarily on colchicine and corticosteroids. NSAIDs have a relatively limited role in this often elderly population and in considering the cardiovascular risk associated with acute CPP flares [12].

A randomized trial compared colchicine (1.5 mg the first day, then 1 mg the next day; $n = 49$) with prednisone (30 mg; $n = 46$) in the treatment of acute CPP arthritis in a hospital setting (COLCHICORT) [13]. Patients were elderly (median age 88 years [IQR 45.2–81.5]) with arthritis predominantly in the knee (48%) and wrist (20%). The mean pain VAS was 68 mm (SD 17). The primary endpoint, change in pain VAS at 24 h, was equivalent in both groups and clinically significant (−36 mm [SD 32] in the colchicine group and −38 mm [SD 23] in the prednisone group). The pain course was similar in both groups up to 7 days after enrollment.

Safety was the main difference between the two groups: 12 (22%) patients in the colchicine group experienced diarrhea compared to 3 (6%) in the prednisone group and there were more non-serious glycemic or blood pressure disturbances in the prednisone group. This excess risk of diarrhea led the authors to recommend the use of prednisone rather than colchicine in the treatment of acute PPC arthritis because of the risk of dehydration in hospitalized elderly patients.

This high-quality randomized trial, one of the first in CPPD, paves the way for future trials, which will undoubtedly be facilitated by the ACR/EULAR 2023 classification criteria, and will help to improve patient management.

Other avenues for treating CPP crystal-induced inflammation are emerging. In particular, a preclinical study targeted monoamine oxidase B (MAO-B), a mitochondrial enzyme involved in ROS production and activation of the NLRP3 inflammasome [14]. Rasagiline and safinamide, irreversible and reversible MAO-B inhibitors, respectively, were tested in two mouse models of microcrystalline inflammation: the air pouch model and the CPP crystal-induced arthritis model. Mice treated with both molecules developed significantly less inflammation after exposure to CPP crystals than control mice, particularly by reducing

ROS generation, inflammasome activation, and production of pro-inflammatory cytokines (such as IL-1 and IL-6). This therapeutic avenue could therefore be explored in the treatment of microcrystalline inflammation, especially as these two molecules are already used clinically in the treatment of Parkinson's disease.

### 3. BCP Crystals

News in the field of BCP crystals in 2023 was mainly characterized by basic research publications. For example, in a preclinical study, BCP crystals did not have a pro-inflammatory effect on human tenocytes but increased the expression of genes associated with tissue remodeling (MMP-1, MMP-3, ADAMTS-4, and TIMP-1) [15]. These results therefore suggest that tendon calcifications directly contribute to the loss of integrity of the tendon matrix by mechanisms independent of inflammation.

Two important studies have investigated the role of BCP crystals in OA. Stassen et al. confirmed that BCP crystals had a pro-catabolic effect on human OA chondrocytes, increasing their production of IL-6, a key cytokine in cartilage degradation during OA [16]. Secretome analysis of chondrocytes exposed to BCP crystals identified several proteins whose expression was increased by the crystals, mainly involved in cartilage extracellular matrix remodeling (especially MMP-1 and TIMP-1), inflammation, and TGF-β signaling. Activation of the TGF-β pathway by BCP crystals is thought to be responsible for IL-6 production by chondrocytes, particularly via the non-canonical signaling pathway TAK-1.

In peripheral blood mononuclear cells (PBMC), BCP crystals potentiated the proinflammatory effects of LPS on IL-1β and IL-8 expression and decreased IL-1Ra expression. Klück et al. identified several SNPs that influence cytokine production by BCP crystals and LPS in PBMC and that may be involved in the co-regulation of IL-1β with other proinflammatory cytokines (including IL-6, IL-8, and IL-1Ra) [17]. A gene of interest identified in this study is *ANO3*, which encodes anoctamin 3, a calcium-activated chloride channel. The inhibition of anoctamin 3 limited IL-1β production by BCP crystals and LPS. Finally, colocalization analysis with signals identified in the largest GWAS of OA identified a locus shared with knee osteoarthritis: *GLIS3* [18]. This suggests that the association of this locus with knee OA is explained by inflammatory mechanisms, specifically an increase in PBMC sensitivity to BCP crystals.

What do these two studies tell us? They reinforce the idea that BCP crystals, which are almost systematically found in the cartilage of OA knees after knee replacement surgery, play an active role in the pathogenesis of osteoarthritis, through pro-catabolic and proinflammatory mechanisms in cartilage and mononuclear cells.

### 4. Osteoarthritis and Cartilage Calcifications

The association between cartilage calcifications and OA has been the subject of several epidemiological studies. Analysis of the MOST study confirmed the association of IA mineralization (detected by CT scan) with age and with Kellgren and Lawrence grade [11]. However, the relationship between the presence of IA calcium deposits and the risk of progression of knee OA remains controversial; epidemiological studies have shown conflicting results [19–21].

In the Osteoarthritis Initiative cohort, Ibad et al. examined the impact of the presence of IA calcifications in the knee and hand (detected by radiography) in 2010 patients with knee OA [22]. The risk of radiographic progression at 8 years was similar in patients with and without IA calcifications. However, in the subgroup of patients under 60 years of age, the presence of IA mineralization increased the risk of radiographic progression (HR = 1.90 [95%CI 1.01–3.60]), particularly in patients with calcifications in the hand (HR = 10.37 [95%CI 3.03–35.46]). This result should be interpreted with caution: in this study, only 46 (2.3%) patients had hand calcifications.

A New Zealand study of 99 patients suggested that the inflammatory phenotype of CPPD influences the risk of OA progression [23]. In this study, the rate of total hip or

knee replacement in patients with acute CPP arthritis was higher than that expected in the general population (RR 2.54 [1.39–4.27]).

The results of these two studies need to be confirmed but they suggest that IA calcifications increase the risk of progression of OA when they are early (<60 years) and systemic and when they induce acute inflammatory flares.

Beyond these very specific cases, which are more likely to be caused by CPP crystals, it is important to remember that the greatest risk factors for cartilage calcification remain age and OA, in which BCP crystals are the most common. One study investigated the role of cellular senescence in the formation of cartilage calcifications in OA [24]. In this study, the authors showed that sortilin expression in articular chondrocytes is increased during OA and is associated with markers of cellular senescence. Sortilin is a transmembrane protein involved in the pathogenesis of vascular calcification and its expression is also increased in articular chondrocytes cultured under calcifying conditions [24]. It may therefore be a novel biomarker of cartilage calcification in osteoarthritis and a novel therapeutic target.

There are still many unanswered questions about the natural history and role of calcifications in cartilage degradation in OA. New imaging techniques with much higher resolution than those routinely available today will certainly help us better understand the prevalence and respective roles of CPP and BCP crystals in the pathophysiology of OA in the coming years [25].

### 5. Conclusions

The tide is turning and significant advances have been made in the field of calcium-containing crystal deposition diseases in 2023. The ACR/EULAR classification criteria and the OMERACT semiquantitative ultrasound scoring system for measuring the extent of CPP deposition in the knee and wrist were the most important tools developed for CPPD and were published in leading journals. The first large-scale randomized controlled trial for the treatment of acute CPP arthritis was also conducted in 2023. Preclinical and epidemiological data have also led to a better understanding of the role of calcium crystals in the pathogenesis of osteoarthritis: in the coming years, the new classification criteria and advanced imaging modalities are expected to facilitate research, further our knowledge in this field, and eventually help us improve the management of patients affected by these common conditions.

**Funding:** This research received no external funding.

**Institutional Review Board Statement:** Not applicable.

**Informed Consent Statement:** Not applicable.

**Data Availability Statement:** Not applicable.

**Conflicts of Interest:** The authors declare no conflicts of interest.

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
