# Peer review of "Calcium Pyrophosphate and Basic Calcium Phosphate Crystal Arthritis: 2023 in Review"

_2813-4583, doi:10.3390/gucdd2020010_

Round 1
Reviewer 1 Report
Comments and Suggestions for Authors
Dear editor,
Please find below my comments on “Calcium pyrophosphate and basic calcium phosphate crystal arthritis: 2023 in review”.
In this review, the authors proposed to report recent advances in the field of CCP (calcium pyrophosphate deposition) and BCP (Basic Calcium Phosphate) deposition diseases. The review focused on the new ACR/EULAR classification criteria, new imaging criteria and advances in therapeutic strategies.
The review is well written and give a nice overview of recent findings.
I have two minor comments:
- page 2, line 48-49 : the authors should add a few more details on the relative criteria used in the classification system in the absence of the two sufficient criteria.
- It would have been interesting to have more information about perspectives in the field : recruiting clinical trials for example. However, looking at Clinicaltrial.gov, there aren't any so, as the authors did, we hope that the new clinical criteria will lead to the development of future clinical studies in CPPD.
Author Response
We would like to thank the reviewer for their positive feedback on our manuscript.
Comment 1 : page 2, line 48-49 : the authors should add a few more details on the relative criteria used in the classification system in the absence of the two sufficient criteria.
Response 1 : we have modified the paragraph as requested to provide more detail on the relative classification criteria.
Comment 2 : It would have been interesting to have more information about perspectives in the field : recruiting clinical trials for example. However, looking at Clinicaltrial.gov, there aren't any so, as the authors did, we hope that the new clinical criteria will lead to the development of future clinical studies in CPPD.
Response 2 : We agree with the reviewer. We have explained how these criteria can be used to facilitate clinical trials or the creation of epidemiologic cohorts in our revised manuscript.
Reviewer 2 Report
Comments and Suggestions for Authors
This is a great review summarizing the current knowledge on articular calcification. The article highlights the novel EULAR criteria for Calcium Pyrophosphate Deposition Disease. Furthermore, it summarizes the efforts in imaging of intra articular calcium crystals and the novel developments in the field. Importantly, it gives an overview on the current cohort studies and summarizes the current results from these cohort studies. Furthermore, I suggest to include sortilin and cartilage calcification to be added to the molecular pathways section.
Author Response
We would like to thank the reviewer for their positive feedback on our review, much appreciated. We have incorporated their suggestions into our manuscript in the section on cartilage calcifications in OA: l. 206-219 in the revised manuscript.